# GeoSense: Internalizing Geometric Necessity Perception for Multimodal Reasoning

**Ruiheng Liu** [* 1]  **Haihong Hao** [* 1]  **Mingfei Han** [* 1 2]  **Xin Gu** [3]  **Kecheng Zhang** [1]  **Changlin Li** [4]  **Xiaojun Chang** [1 2]

## Abstract

Advancing towards artificial superintelligence requires rich and intelligent perceptual capabilities. A critical frontier in this pursuit is overcoming the limited spatial understanding of Multimodal Large Language Models (MLLMs), where geometry information is essential. Existing methods often address this by rigidly injecting geometric signals into every input, while ignoring their necessity and adding computation overhead. Contrary to this paradigm, our framework endows the model with an awareness of perceptual insufficiency, empowering it to autonomously engage geometric features in reasoning when 2D cues are deemed insufficient. To achieve this, we first introduce an independent geometry input channel to the model architecture and conduct alignment training, enabling the effective utilization of geometric features. Subsequently, to endow the model with perceptual awareness, we curate a dedicated spatial-aware supervised fine-tuning dataset. This serves to activate the model's latent internal cues, empowering it to autonomously determine the necessity of geometric information. Experiments across multiple spatial reasoning benchmarks validate this approach, demonstrating significant spatial gains without compromising 2D visual reasoning capabilities, offering a path toward more robust, efficient and self-aware multi-modal intelligence.

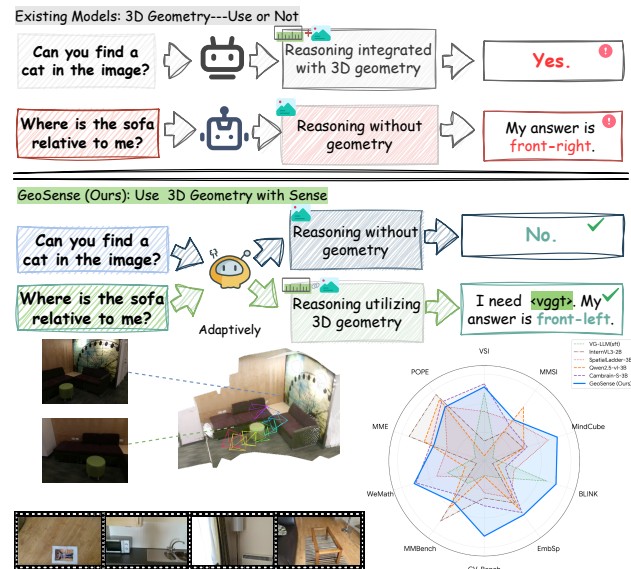

*Figure 1.* **Adaptive Geometric Reasoning with GeoSense.** (Top) Existing MLLMs typically adopt a static approach to 3D geometry, either ignoring it or rigidly fusing it, which leads to confusion in general tasks or failures in spatial reasoning. (Bottom) GeoSense introduces an adaptive mechanism (*Use with Sense*) that requests geometric features only when necessary. As shown in the radar chart, this flexibility allows GeoSense to achieve SOTA performance across both general visual benchmarks (*e.g.*, MM-Bench (Liu et al., 2024b), WeMath (Qiao et al., 2024)) and spatial reasoning tasks (*e.g.*, VSI-Bench (Yang et al., 2025a) and Mind-Cube (Yin et al., 2025)).

---

[*]Equal contribution [1]University of Science and Technology of China, Anhui, China [2]Mohamed bin Zayed University of Artificial Intelligence, Abu Dhabi, UAE [3]University of Chinese Academy of Sciences, Beijing, China [4]Stanford University, CA, USA. Correspondence to: Xiaojun Chang <xjchang@ustc.edu.cn>.

*Proceedings of the 43$^{rd}$ International Conference on Machine Learning*, Seoul, South Korea. PMLR 306, 2026. Copyright 2026 by the author(s).

## 1. Introduction

Multimodal Large Language Models (MLLMs) have revolutionized domains such as autonomous driving, embodied AI, and industrial robotics (Xie et al., 2025; Li et al., 2024b; Szot et al., 2025; Ji et al., 2025). However, as these models transition to real-world deployment, spatial reasoning has emerged as a critical bottleneck that affects both reliability and decision accuracy. While recent works (Yang et al., 2025a; Liu et al., 2025b; Daxberger et al., 2025; Wu et al., 2025a; Zheng et al., 2025) have shown that incorporating geometric information significantly enhances spatial understanding, the effective integration of these cues remains a challenge.

Due to the scarcity of native 3D data such as depth maps and point clouds (Wu et al., 2025a), research has shifted toward reconstructing 3D representations from 2D visual inputs using foundation models like VGGT (Wang et al., 2025a) and MoGe-2 (Wang et al., 2025c)). Enabled by these 3D foundation models, recent approaches have begun injecting geometry-encoded embeddings derived from 2D visual inputs directly into the reasoning pipeline (Zheng et al., 2025; Wu et al., 2025a). For example, Spatial-MLLM (Wu et al., 2025a) projects the geometric features from VGGT (Wang et al., 2025a) into the pretrained vision-language space to provide 3D cues during inference, fostering an implicit internalized 3D awareness. However, these approaches typically follow a tightly coupled design that treats 3D geometric information as a mandatory input, regardless of the specific task requirements. This assumes geometry as a universal necessity, yet in practice, its utility is highly context-dependent. For instance, while depth cues are vital for spatial navigation, they serve as irrelevant noise for non-spatial tasks such as table OCR or plane geometry. This forced activation of geometric signals can degrade general reasoning capabilities by introducing unnecessary complexity and degrades general reasoning performance.

We identify this limitation as the perception information gap: the discrepancy between the information a model perceives from standard 2D inputs, and the specific context and domain knowledge required to solve complex spatial reasoning problems. Current MLLMs lack the internal awareness to recognize this gap or assess their own information requirements. To address this, we introduce *GeoSense*, a framework that internalizes the perception of geometric necessity. Our approach endows the model with the intrinsic capability to autonomously decide whether to integrate 3D geometric cues based solely on the input content and query, without requiring explicit spatial instructions.

Specifically, our framework training consists of two stages. First is the geometry alignment. We treat geometric information as a standalone modality via an independent geometry input channel rather than element-wise addition (Zheng et al., 2025; Wu et al., 2025a) to 2D features. This allows geometric features to serve as an on-demand resource rather than a mandatory burden. The second is spatial perception tuning. We curate a model-adaptive, spatially-aware dataset by evaluating the model's own performance discrepancies with and without geometric features, and reformulate these variances into training signals. This allows the model to learn its own empirical priors, superseding rigid, human-defined rules.

As a result, our model dynamically modulates its reliance on geometry input, achieving superior spatial reasoning while preserving general-purpose visual intelligence. Releasing dynamically the burden of an additional spatial encoder, our

approach is particularly suitable for small scale MLLMs deployed on edge devices. In summary, our contributions are summarized as follows.

- We propose a two-stage training framework that enables the model to autonomously select reasoning pathway based on context-specific information demands.

- We construct a model-adaptive data curation pipeline that captures intrinsic empirical priors of MLLMs, internalizing the perception of geometry necessity without relying on hand-crafted rule systems.

- We show that our method significantly improves spatial reasoning performance while maintaining robust general-purpose reasoning across diverse benchmarks.

**Conflict of Interest Disclosure.** The authors declare no financial conflicts of interest related to this work.

## 2. Related Work

### 2.1. MLLMs for Visual Scene Understanding

The evolution of Multimodal Large Language Models (MLLMs) has significantly advanced visual scene understanding. Representative approaches, such as BLIP-2 (Li et al., 2023a) and LLaVA (Liu et al., 2023; 2024a), align visual encoders with LLMs via lightweight projectors, enabling robust 2D perception. Recent state-of-the-art models like Qwen2.5-VL (Bai et al., 2025) and InternVL-3 (Zhu et al., 2025b) further push performance boundaries through high-resolution inputs and dynamic token allocation. However, these 2D-native architectures inherently lack explicit geometric representations, leading to systematic failures in tasks requiring structural consistency and perspective manipulation (Zhu et al., 2024; Liu et al., 2025a).

To bridge this gap, research has branched into 2.5D strategies injecting depth cues (Zhu et al., 2025a; Huang et al., 2025) and 3D-native models processing point clouds (Xu et al., 2024; Mao et al., 2025). While effective, 3D-native methods often entangle appearance and geometry at the input level. More recently, hybrid approaches have attempted to fuse features from 3D foundation models directly into 2D visual embeddings (Wu et al., 2025a; Zheng et al., 2025; Chen et al., 2025). Unlike these fusion-based methods, which risk compromising the model's original planar understanding, our approach treats 3D features as a distinct, independent input modality, ensuring that 2D visual reasoning remains unaffected.

### 2.2. Visual Spatial Intelligence

Spatial intelligence encompasses relational reasoning (*e.g.*, depth, topology) and viewpoint transformation (Yang et al.,

2025a; Song et al., 2025). Despite excelling in general visual tasks, MLLMs often struggle with these spatial dimensions due to the scarcity of 3D annotated data (Hudson & Manning, 2019; Li et al., 2024a). Early solutions relied on synthetic datasets to teach basic spatial concepts (Chen et al., 2024a; Cheng et al., 2024), but these failed to capture dynamic real-world complexities.

With the advent of powerful geometric estimation models (Wang et al., 2025a;c), recent works utilize predicted 3D priors to enhance MLLMs without costly 3D annotations. Systems like SpatialMLLM (Wu et al., 2025a) and VGLLM (Zheng et al., 2025) integrate these priors into the visual latent space. However, such indiscriminate incorporation forces the model to process 3D information even when unnecessary, which has been shown to impair performance on general reasoning tasks like mathematics or OCR (Foroutan et al., 2025; Wang et al., 2025b). In contrast, our work introduces an adaptive mechanism that selectively incorporates 3D information based on task necessity, thereby enhancing spatial understanding while preserving the model's general reasoning capabilities acquired during pretraining.

## 3. Method

We propose *GeoSense*, a multimodal framework designed to adaptively integrate 3D cues based on its internal geometric necessity perception. Unlike conventional approaches that rigidly fuse geometric features, GeoSense employs a decoupled framework where geometric information serves as an on-demand resource. This capability is realized through a novel request mechanism learned via a two-stage supervised fine-tuning strategy, allowing the model to autonomously determine when geometric injection is strictly necessary.

### 3.1. Geometry Cues Are Not Universally Beneficial

To realize adaptive spatial perception, we must first determine *when* geometry is actually needed. However, defining a universal heuristic for this is intractable. While a photorealistic image may only require 2D object counting, a simple line drawing might demand complex 3D depth reasoning. Much like human spatial awareness, the necessity of geometry is highly context-dependent and rooted in implicit intuition rather than rigid rules.

Hypothesizing that MLLMs can develop similar latent priors, we aimed to empower it to articulate its own needs rather than relying on hand-crafted rules. To validate this, we evaluated a standard VG-LLM (Zheng et al., 2025) on a mixture of spatial reasoning (Wu et al., 2025a) and general VQA dataset (Zhang et al., 2024) under two conditions: (*i*) with geometry embeddings injected, and (*ii*) with geometry inputs suppressed via zero-padding. Quantitative analysis of approximately 700k samples revealed a critical insight. For ∼67% of the samples, the model was robust regardless of geometry usage, while ∼25% failed under both conditions (likely due to base capacity limits). The remaining ∼8% highlighted the "double-edged sword" nature of geometric injection: while 5% of samples *required* 3D features for a correct response, the remaining 3% actually suffered performance degradation when geometric features were added.

This counter-intuitive finding, that geometry can act as noise, was consistent across both general benchmarks (where geometry suppression improved performance) and spatial benchmarks (where geometry inclusion was vital), comparing between VG-LLM (Zheng et al., 2025) and Qwen2.5-VL-3B (Bai et al., 2025), as shown in Table 2. *Crucially, this issue is exacerbated by scale.* Enlarging the training scale (from 385k to 940k) and increasing spatial data under a rigid integration paradigm (*i.e.*, VG-LLM) further degrades the performance on general visual benchmarks, leading to severe hallucinations (POPE (Li et al., 2023b) drops from 86.9 to 74.2) and cross-modal confusion (average score drops from 52.0 to 48.2). These findings confirm that rigid integration is suboptimal, establishing the empirical foundation for our adaptive data construction and necessitating the adaptive framework with our GeoSense.

### 3.2. Independent Geometry Adaptation

To achieve autonomous spatial awareness without compromising native 2D perception, it requires a decoupled input architecture. Unlike prior approaches that fuse geometric features via element-wise summation (Zheng et al., 2025; Wu et al., 2025a), we treat 3D geometry as an independent, on-demand modality. This design ensures that the high-resolution 2D visual stream, responsible for general recognition, remains unpolluted by geometric embeddings.

**Model Architecture.** We utilize the visual encoder of Qwen2.5-VL-3B (Bai et al., 2025) and VGGT (Wang et al., 2025a) as our 2D and geometric encoders. Upon receiving visual input, both encoders extract features which are projected via dedicated MLP modules to a shared embedding space. These projected features are then populated into specific placeholders, `<|vision_pad|>` and `<|vggt_pad|>`, to construct a comprehensive multimodal prompt containing discrete 2D, geometric, and textual components.

**Geometry sequence formulation.** Formally, given a visual input $\mathcal{V}$, the 3D encoder $\mathcal{E}_{\text{geo}}$ extracts structured geometric features:

$$F_{\text{geo}} = \mathcal{E}_{\text{geo}}(\mathcal{V}) \in \mathbb{R}^{T \times D_{\text{3D}}}, \tag{1}$$

where $T$ denotes the sequence length and $D_{\text{3D}}$ the raw feature dimension. To align these with the MLLM's embedding

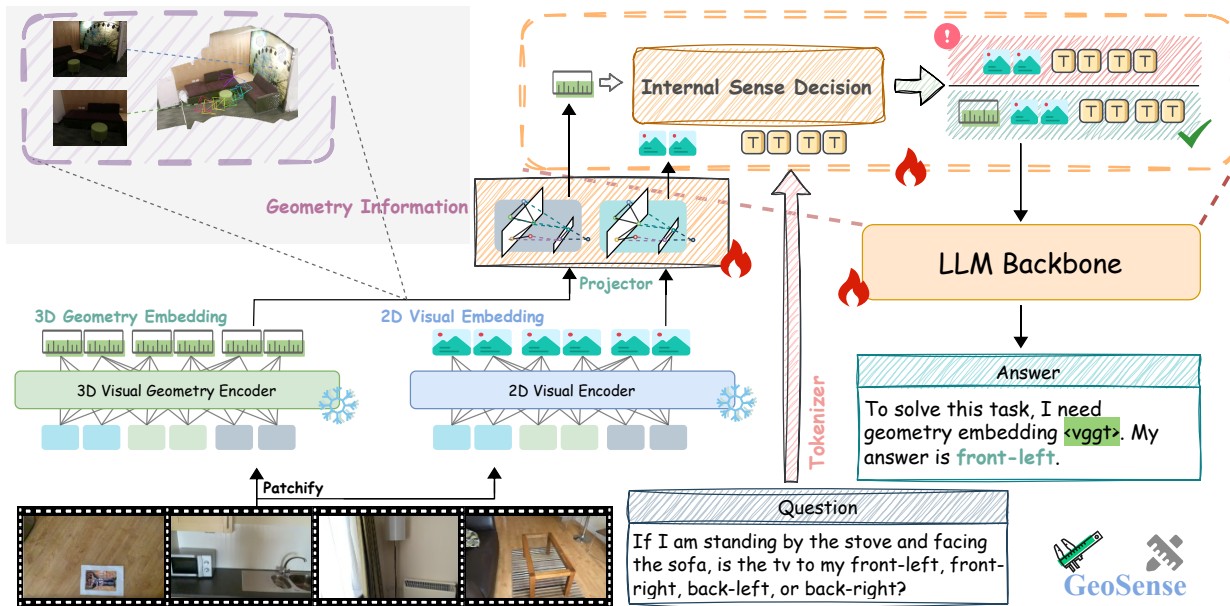

*Figure 2.* **Architectural Overview of GeoSense.** We integrate a 3D visual geometry encoder alongside a standard 2D visual encoder, both of which are kept frozen to preserve pretrained representations. Dedicated projection layers map these features into a unified embedding space for the LLM backbone. During inference, the model dynamically makes an "Internal Sense Decision" based on the 2D visual and textual prompt. If the latent state triggers a geometry request (*e.g.*, via the `<vggt>` token), 3D embeddings are concatenated to the sequence for a second re-inference pass. During training, only the projection layers and the LLM backbone are optimized.

space ($D_{MM}$), we apply a learnable linear projection $W^{3D}_{proj}$:

$$T_{geo} = W^{3D}_{proj} F_{geo} \in \mathbb{R}^{T \times D_{MM}}. \quad (2)$$

Finally, the projected geometry tokens are serialized as an independent segment delimited by boundary tokens ($E_{start\_g}, E_{end\_g}$) and concatenated with text ($T_{text}$) and 2D visual tokens ($T_{vision}$):

$$H_{input} = \begin{bmatrix} T_{text} \oplus T_{vision} \oplus E_{start\_g} \oplus T_{geo} \oplus E_{end\_g} \end{bmatrix}. \quad (3)$$

### 3.3. Activating Internal Spatial Awareness

We employ a two-stage training strategy to progressively instill adaptive spatial capabilities, transitioning the model from basic modality alignment to autonomous geometric reasoning, as shown in Figure 2.

**Geometric Feature Alignment.** As shown in Fig. 2, the initial phase projects 2D and 3D features into a unified embedding space via dual projection modules. During this stage, the pre-trained feature extractors ($\mathcal{E}_{vision}$ and $\mathcal{E}_{geo}$) are frozen to preserve their representational integrity, while the LLM backbone, the 2D/3D projectors ($W^{2D}_{proj}, W^{3D}_{proj}$), and boundary tokens are optimized. This ensures that 3D tokens are semantically aligned with the existing visual-textual space. To provide a robust foundation for this alignment, we utilize a mixture of temporal and spatial grounding datasets, including LLaVA-Hound-64k (Zhang et al., 2024) and Spar-234k (Wu et al., 2025a).

**Spatial-Aware Supervised Fine-Tuning.** Building upon the aligned feature space, this stage evolves the model into an active perceiver capable of making "Internal Sense Decision". Rather than treating geometry input as mandatory, the model is trained to function as an adaptive cognitive gate. As shown by the branching paths in Figure 2, the model learns to dynamically evaluate visual input and the task intent. For tasks requiring high spatial precision, the model is trained to trigger a signal, *i.e.*, `<vggt>`. It acts as a request for geometric embeddings; if the signal is not triggered, the model suppresses the channel to maintain 2D visual reasoning purity. This mechanism ensures that geometric processing is invoked only when necessary, effectively mitigating the noise and computational overhead associated with rigid geometry integration.

## 4. Data Curation

To extract and internalize the model's intrinsic empirical priors, we construct a comprehensive hybrid dataset derived from VSI-590k (Yang et al., 2025d), SophiaVL-R1-130k (Fan et al., 2025), and Mantis-Instruct (Jiang et al., 2024). We perform dual-condition inference on these samples, generating predictions under two controlled settings: a 2D-only setting and a geometry-augmented setting with 3D geometric features. For both settings, we use the same decoding configuration and answer parser to compare the model prediction against the original ground-truth answer.

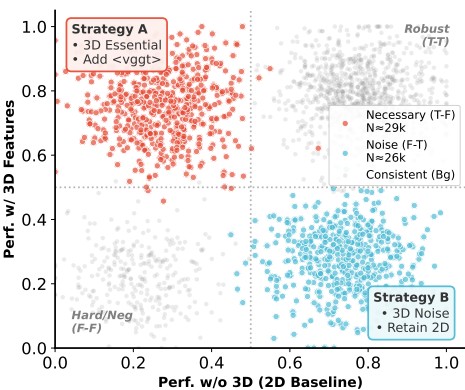

*Figure 3.* Sample distribution of the perception dataset by training objective. Consistent refers to samples with prediction invariance to 3D features, kept as is. Strategy A and Strategy B represent data where 3D geometric cues are essential and where they should be taken as noise, respectively.

Each raw sample is therefore assigned to one of four quadrants according to the correctness of the two predictions: True-True (T-T), True-False (T-F), False-True (F-T), and False-False (F-F), as illustrated in Figure 3. Ambiguous, invalid, or unparsable samples are filtered out before rewriting, so the retained supervision is derived from reliable differences between the two inference conditions.

Our experiments reveal a critical discrepancy in existing dataset. Even within VSI-590K, which is tailored for indoor spatial reasoning, approximately 25,000 samples exhibit performance divergence when 3D features are introduced. To address this, we implement a differentiated data reconstruction strategy:

- **Necessary Geometry (Strategy A):** For *T-F* samples where 3D cues are essential, we restructure the data into a two-turn dialogue. The first turn prompts the model to generate a Chain-of-Thought (CoT) that concludes with the `<vggt>` trigger signal, explicitly stating the need for geometric integration. The second turn then provides the final answer.

- **Geometry as Noise (Strategy B):** For *F-T* samples where 2D features are superior, we retain the original labels while specifically training the model to suppress the trigger signal, thereby relying solely on 2D visual context.

As summarized in Table 1, this methodology yields 117k task-aware training samples. This strategy is methodologically significant as it decouples scene context from task types. By ensuring that identical scenes can trigger opposite supervision signals (activation vs. suppression of geometry input), we effectively mitigate shortcut learning. The model

*Table 1.* Composition of the Perception Tuning Data.

| Dataset Source | Subset / Type | Data Size |
|---|---|---|
| VSI-590K | Sampled Subset | 55K |
| SophiaVL-R1 | Chart | 5K |
| | General | 15K |
| | Knowledge | 3K |
| Mantis-Instruct | LRV-Multi | 13K |
| | NLVR2 | 16K |
| Llava-Hound-64K | Sampled Subset | 10K |

is thus compelled to make "Internal Sense Decisions" based on a genuine perception of information necessity rather than memorizing background templates.

## 5. Experiments

In this section, we first detail the experimental setup and implementation (Section 5.1), followed by a comparison against SOTA MLLMs on spatial reasoning and general visual benchmarks (Section 5.2). Finally, we present ablation studies (Section 5.3) and case study of model inference Section 5.6.

**Implementation details.** Our framework is built upon the Qwen2.5-VL-3B (Bai et al., 2025), integrated with VGGT-1B (Wang et al., 2025a) as the 3D foundation model. We utilize a two-stage training process: *Alignment Phase:* The model is fine-tuned for one epoch on a mixed alignment dataset as stated in Section 4, with a batch size of 32. Each training batch is randomly sampled from a single source to ensure balanced data distribution. We use the Adam optimizer (Kingma & Ba) with a learning rate of $1 \times 10^{-6}$ and a 0.03 warmup ratio. *Spatial-Aware Phase:* We maintain the initial hyperparameters but increase the batch size to 96. Throughout both stages, the visual and 3D encoders remain frozen, and only the LLM backbone and projection layers are optimized. Experiments were conducted on 8 NVIDIA A100 (80GB) GPUs, requiring 14 hours for alignment and 20 hours for spatial-aware fine-tuning.

### 5.1. Experimental Setup

To provide a rigorous evaluation of spatial intelligence and general capabilities, we benchmark our approach against baseline models and several representative state-of-the-art models across a comprehensive suite of 10 diverse datasets.

**Baseline Models.** We compare against two categories of baselines: *Specialized Spatial Models* and *General Multimodal Baselines*. For specialized models, we evaluate SpatialLadder-3B (Li et al., 2025), which employs a progressive curriculum training with Group Relative Policy Optimization (GRPO), and ViLaSR-7B (Wu et al., 2025b), which integrates an "Interwoven Thinking and Visual Drawing" mechanism for embodied reasoning. In the video

domain, SpaceR-sft-7B (Ouyang et al., 2025) targets spatiotemporal geometry, while Cambrian-S-3B (Yang et al., 2025d) utilizes a surprise-driven memory mechanism for spatial supersensing. We also include VG-LLM, which injects explicit 3D vision geometry priors, and VST-3B-SFT (Yang et al., 2025b), a data-centric approach using visual spatial tuning. For general baselines, we utilize Qwen2.5-VL (7B/3B) (Bai et al., 2025), featuring Naive Dynamic Resolution and absolute time encoding, and InternVL3-2B (Chen et al., 2024b), which demonstrates the efficiency of native multi-modal pre-training with Variable Vision Positional Encoding (V2PE).

**Spatial Reasoning Benchmarks.** To rigorously assess spatial intelligence, we employ six benchmarks across diverse dimensions. VSI-Bench (Yang et al., 2025a) is used for holistic 3D understanding, including configurational and measurement tasks. For multi-view and mental simulation capabilities, we utilize MMSI-Bench (Yang et al., 2025c) and MindCube-Tiny (Yin et al., 2025), which require constructing mental models from limited views. Low-level geometric perception is evaluated using BLINK (Fu et al., 2024), which resists language mediation, and the 3D subset of CV-Bench (Tong et al., 2024) for depth and occlusion reasoning. Additionally, EmbSpatial (Du et al., 2024) is employed to test egocentric spatial understanding for embodied tasks. We report the arithmetic mean of these datasets as the *Spatial Avg.*.

**General Visual Benchmarks.** To ensure that spatial specialization does not compromise general capabilities, we evaluate models on four established benchmarks. MMBench (Liu et al., 2024b) provides a comprehensive assessment using CircularEval to mitigate bias. We use MME (Fu et al., 2025) (Perception Score) to measure recognition breadth and POPE (Li et al., 2023b) to strictly detect object hallucinations. Furthermore, WeMath (Qiao et al., 2024) is included as a proxy for high-level logical reasoning and geometric abstraction. The *General Avg.* aggregates these scores to reflect overall model robustness.

## 5.2. Performance Comparison

As demonstrated in Table 2, our proposed model significantly outperforms the baseline Qwen2.5-VL-3B (Bai et al., 2025) on spatial reasoning benchmarks under comparable training data scales. Notably, our model achieves competitive spatial reasoning performance relative to the substantially larger Qwen2.5-VL-7B (Bai et al., 2025). Concurrently, our model performs well for general visual reasoning tasks by dynamically reverting to standard 2D and textual embedding combinations. The system effectively mitigates cross-modal interference typically caused by redundant geometric features in non-spatial contexts.

**Performance on spatial reasoning.** Specifically, our model achieves superior comprehensive performance by internalizing the perception of spatial feature requirements. For instance, in the BLINK (Fu et al., 2024) benchmark, which comprises diverse spatial subtasks with varying dependencies on geometric relations, our model attains the highest overall score. This reinforces our objective: enabling the model to adaptively utilize geometric features based on task demands, thus enhancing spatial reasoning while preserving general inference capabilities. Furthermore, to isolate the impact of training data, we compare our model with the fine-tuned baseline (marked with *) using identical datasets. On the EmbSpatial (Du et al., 2024) subset, while the VG-LLM model (which consistently introduces 3D features) exhibits a performance decline after fine-tuning, our approach effectively improves spatial reasoning through perception-driven sample selection. This underscores that our gains derive from the efficacy of the mechanism rather than merely data quality.

**Performance on general visual reasoning.** For general visual tasks, Our model tends to suppress redundant interference, with geometric features being triggered for only approximately 3% of samples. Surprisingly, our model also achieves superior performance on the WeMath (Qiao et al., 2024) dataset, which could be attributed to a significant portion of the test set involving spatial-geometric mathematical problems. Compared to models that blindly incorporate geometric features, our approach yields a significant improvement, demonstrating that an activated perceptual capability allows the model to retain essential spatial imagination for complex problem-solving (*e.g.*, math and physics) even without explicit external geometric inputs.

## 5.3. Ablation Study

**Comparison of geometry injection scheme.** We conducted an ablation study using identical training data to evaluate three VGGT (Wang et al., 2025a) integration schemes: (1) w/o VGGT (baseline), (2) Visual Fusion (element-wise addition), and (3) our proposed Adaptive Selection. Detailed comparisons were performed on two mainstream spatial reasoning benchmarks: VSI-Bench (Yang et al., 2025a) and MindCube-Tiny (Yin et al., 2025). Experimental results demonstrate that our adaptive scheme activated VGGT features as auxiliary input tokens for an average of 35.68% of samples. Notably, the activation rate is 43.7% on VSI-Bench and 27.58% on MindCube-Tiny. This variance underscores the flexibility of our approach. For instance, in the Among category of MindCube, the task requires spatial logic (locating cues based on descriptions) rather than spatial reconstruction. Our model correctly identified that geometric features was unnecessary for these samples, thereby avoiding redundant computation.

*Table 2.* Performance comparison on Spatial and General benchmarks. We report average scores for each benchmark collection, seperate benchmarks, and a total average for ranking. For spatial reasoning benchmarks, we include VSI-Bench (Yang et al., 2025a), MMSI (Yang et al., 2025c), MindCube (abbr. as MC) (Yin et al., 2025), BLINK (Fu et al., 2024), EmbSpatial (Du et al., 2024) and CV-Bench(abbr. as CVB) (Tong et al., 2024) for comprehensive spatial reasoning evaluation. For general benchmarks, we include MMBench (Liu et al., 2024b), MME (Fu et al., 2025), POPE (Li et al., 2023b), and WeMath (Qiao et al., 2024) to evaluate the general multimodal reasoning capability. Note that we evaluate the 3D subset of CV-Bench and the MME perception score. * denotes fine-tuning on the same data as our model used.

| Model | FT-Data | Spatial Reasoning Benchmarks | | | | | | | General Benchmarks | | | | | Overall | |
| | | Avg. | VSI-Bench | MMSI | MC-Tiny | BLINK | EmbSpatial | CVB(3D) | Avg. | MMBench | MME (P) | POPE | WeMath | Avg. | Rank |
|---|---|---|---|---|---|---|---|---|---|---|---|---|---|---|---|
| Qwen2.5-VL-3B (Bai et al., 2025) | - | 43.4 | 27.0 | 28.6 | 37.6 | 33.1 | 62.3 | 71.8 | 53.3 | 76.6 | 1526.1 | 87.5 | 25.8 | 48.3 | 11 |
| Qwen2.5-VL-7B (Bai et al., 2025) | - | 50.5 | 32.3 | 26.8 | 36.0 | 55.9 | 71.8 | 80.1 | 57.8 | 82.6 | 1693.9 | 87.8 | 33.8 | 54.1 | 3 |
| InternVL3-2B (Chen et al., 2024b) | - | 43.6 | 33.0 | 26.5 | 37.5 | 30.0 | 60.1 | 74.3 | 54.1 | 78.6 | 1610.2 | 88.9 | 22.4 | 48.8 | 9 |
| ViLASR-7B (Wu et al., 2025b) | 73K | _51.0_ | 44.6 | 30.2 | 35.1 | 51.4 | 67.3 | 77.2 | 54.5 | 80.8 | 1634.4 | 84.8 | 25.3 | 52.8 | 4 |
| SpaceR-sft-7B (Ouyang et al., 2025) | 151K | 49.9 | 41.6 | 27.4 | 38.0 | 49.6 | 66.9 | 75.7 | 58.9 | 82.8 | 1688.1 | 88.0 | 39.0 | 54.4 | 2 |
| SpatialLadder-3B (Li et al., 2025) | 26K | 48.5 | 44.9 | 27.4 | 43.5 | 43.0 | 58.2 | 73.7 | 52.5 | 72.3 | 1403.2 | 85.5 | 34.4 | 50.5 | 7 |
| Cambrian-S-3B (Yang et al., 2025d) | 10M | 49.6 | 56.1 | 27.0 | 38.4 | 37.7 | 63.5 | 75.2 | **55.4** | 74.9 | 1490.8 | 86.8 | 40.7 | 52.5 | 5 |
| VG-LLM (Zheng et al., 2025) | 385k | 49.7 | 47.3 | 27.6 | 31.2 | 48.9 | 64.2 | 79.1 | 52.0 | 75.4 | 1441.6 | 86.9 | 25.6 | 50.9 | 6 |
| VG-LLM* (Zheng et al., 2025) | 940k | 48.8 | 52.3 | 25.2 | 30.1 | 62.8 | 50.9 | 71.7 | 48.2 | 73.2 | 1248.5 | 74.2 | 31.1 | 48.5 | 10 |
| Qwen2.5-VL-3B* (Bai et al., 2025) | 940k | 49.2 | 48.1 | 24.2 | 41.5 | 63.1 | 50.4 | 67.9 | 51.0 | 71.0 | 1301.2 | 85.2 | 33.8 | 50.1 | 8 |
| **GeoSense** | 940K | **56.6** | 54.9 | 27.5 | 45.1 | 68.5 | 64.3 | 78.7 | _55.2_ | 75.9 | 1473.7 | 85.4 | 41.2 | **55.9** | 1 |

*Table 3.* Ablation study on different geometry injection strategy. We compare the baseline against the additive fusion manner (VG-LLM) and our adaptive independent token insertion approach (GeoSense). All variants are trained with the same data to ensure fair comparison.

| Model | Injection | VSI-Bench | | | | | | | | MindCube-Tiny | | |
| | | Avg. | Obj. Count | Obj. Size | Room Size | Rel. Dir. | Route Plan | Appr. Order | Rel. Dist. | Avg. | Rotation | Among | Around |
|---|---|---|---|---|---|---|---|---|---|---|---|---|---|
| Qwen2.5-vl-3B | 0.00% | 48.07 | 63.45 | 58.45 | 41.94 | 46.02 | 31.95 | 59.87 | 49.43 | 35.67 | **34.50** | 41.50 | 27.50 |
| + fusion VGGT (VG-LLM) | 100.00% | 52.34 | 64.67 | 63.96 | 39.06 | **61.61** | **35.56** | 61.00 | **54.78** | 30.58 | 34.00 | 30.00 | 29.75 |
| + alternative VGGT (GeoSense) | 35.68% | **54.86** | **69.43** | **66.64** | **58.26** | 52.85 | 34.02 | **71.52** | 53.66 | **45.08** | 31.50 | **49.17** | **45.75** |

Specifically, our model outperformed the other two schemes on the majority of VSI-Bench (Yang et al., 2025a) tasks. A notable exception is the Relative Direction task, where our model lagged behind the Visual Fusion baseline. We attribute this to VGGT feature for multi-scene contexts. Treating VGGT features for dense scene as an independent modality token appears to amplify the signal-to-noise ratio disadvantage compared to feature-level fusion. Nevertheless, the performance gain over the "w/o VGGT" baseline confirms the validity of incorporating geometric priors.

In contrast, results on MindCube revealed distinct behavior. For the Rotation task, all schemes utilizing VGGT exhibited performance degradation. This is likely due to the data characteristics (wide-angle views with only 2-4 images), which hinder the effective construction of global representations by VGGT. Interestingly, the Visual Fusion scheme performed slightly better than ours in this specific failure case. We hypothesize that fusing VGGT with strong 2D features implicitly dilutes the negative impact of the low-quality geometric representation, whereas our approach exposes the model more directly to the suboptimal signals.

**Effectiveness of components.** To rigorously validate the efficacy of each component within our proposed framework,

we conducted a comprehensive ablation study comparing the model's performance trajectories across different training stages. Specifically, we evaluated three variations: (1) The Initialized Model, which integrates the pre-trained weights of Qwen2.5-VL-3B (Bai et al., 2025) and VGGT-1B (Wang et al., 2025a), while notably initializing the projection layer with weights from a state-of-the-art model to map geometric features into the 2D visual embedding space; (2) The Alignment-Tuned Model; and (3) the final Perception-Tuned Model. We benchmarked these variants against the vanilla Qwen2.5-VL-3B baseline across both spatial and general visual tasks.

**Confidence score of trigger token under difference setting.** As shown in the Table 4, spatial reasoning tasks exhibit a consistent monotonic improvement, with the average score surging from 51.0 to 63.4. Notably, on the challenging BLINK benchmark, our method achieves a remarkable gain (+35.4) compared to the baseline. Conversely, general visual reasoning tasks display a "dip-then-recover" trajectory. We attribute the Initial model's relative robustness to the pre-trained projection layer, which maps geometric embeddings into a pseudo-2D visual space, effectively treating VGGT inputs as implicit visual features. However, the subsequent introduction of special tokens during Align. SFT disrupts

*Table 4.* Ablation Study on baseline and our proposed variants (GeoSense-4B) with different training stages. MME scores are scaled by 28 for average calculation. **Bold** denotes the best performance. Underline denotes the second-highest performance.

| # | Data | Model | Spatial Tasks | | | | General Tasks | | | Avg |
|---|------|-------|---------------|--|--|--|---------------|--|--|-----|
| | | | VSI-Bench | MindCube | BLINK | CV-Bench | MM-Bench | MME | MM-Star | |
| 1 | ∅ | Qwen2.5-VL-3B | 27.0 | 37.6 | 33.1 | 71.8 | **76.6** | **2104** | **56.2** | 51.0 |
| 2 | Initial | GeoSense-4B | 42.8 | 37.8 | 58.9 | 74.9 | 75.2 | 1974 | 51.6 | 58.8 |
| 3 | Align. SFT | GeoSense-4B | 53.9 | 44.4 | 61.9 | 77.7 | 72.2 | 1803 | 45.5 | 60.0 |
| 4 | Percept. SFT | GeoSense-4B | **54.9** | **45.7** | **68.5** | **78.7** | 75.9 | 1922 | 51.7 | **63.4** |

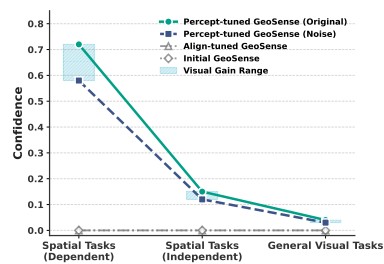

*Figure 4.* Confidence scores of 3D trigger token across spatial and general tasks.

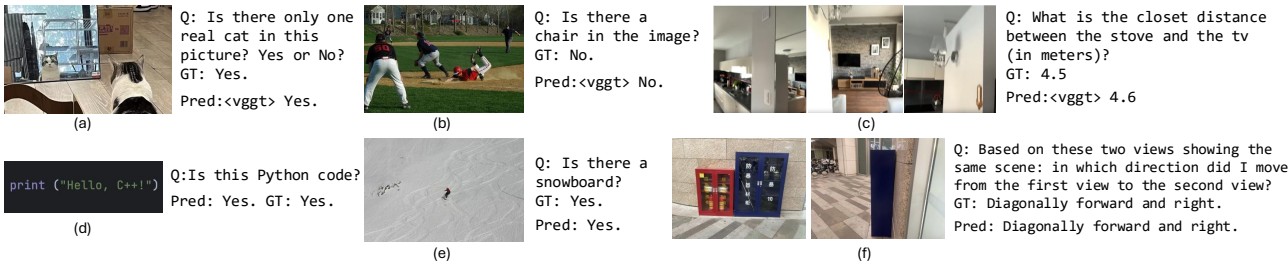

*Figure 5.* **Case study of internal sense decision.** We present representative examples demonstrating how our model adaptively determines whether to trigger 3D geometric features according to the input and task. **(a, b)**: Rare cases where general tasks explicitly demand geometric embedding. **(c)**: Typical activation for spatial reasoning. **(d, e)**: Standard suppression for general visual inputs. **(f)**: Spatial queries solved effectively without geometric triggers.

this implicit mapping, causing a temporary performance degradation (*e.g.*, MME drops to 1803). Crucially, the proposed Percept. SFT reverses this decline. By learning to selectively gate VGGT embeddings, the model mitigates interference, restoring general capabilities (*e.g.*, MME recovers to 1922) while maintaining peak spatial performance. This confirms our design objective: significantly enhancing spatial reasoning while preserving the integrity of general multimodal understanding.

Similarly, we analyzed the confidence contributions of different components regarding the trigger token. As illustrated in Figure 4, we examined the probabilistic behavior of this token across three distinct task categories: geometry-reliant tasks, geometry-agnostic spatial tasks, and general visual tasks. Our results indicate that high confidence scores are exhibited exclusively by the model following perceptual fine-tuning, whereas previous training stages fail to facilitate the triggering of such perceptual signals. To verify that this decision is not driven solely by textual priors, we conducted a control experiment by replacing the input image with random noise. Results demonstrate a significant attenuation in the confidence of the spatial query token under noise conditions. This finding confirms that the model's determination of spatial information requirements is contingent upon a multi-modal synergy between textual prompts and underlying visual semantics, underscoring the effectiveness of our approach in achieving robust, context-dependent

*Table 5.* Comparison of different models on embodied task. **Overall** denotes the aggregate score, while S2/D2 represent static/dynamic 2D sub-task and S3/D3 represent static/dynamic 3D sub-task (reported in %). The best results are highlighted in **bold**, and the second-best results are underlined.

| Model | Overall | S2 | D2 | S3 | D3 |
|-------|---------|-----|-----|-----|-----|
| InternVL3-2B | 29.25 | 21.88 | 27.30 | 25.58 | 39.15 |
| Qwen2.5-VL-3B-Instruct | 36.79 | **37.81** | **44.32** | 27.91 | 38.30 |
| Qwen3-VL-2B-Instruct | 30.44 | 19.69 | 31.62 | 23.02 | 43.62 |
| Qwen3-VL-4B-Instruct | 36.60 | 26.56 | 32.70 | 29.77 | **52.77** |
| SpatialLadder-3B | 36.86 | 34.06 | 34.59 | **33.72** | 43.40 |
| VG-LLM-4B | 31.13 | 34.06 | 36.76 | 17.67 | 37.02 |
| VST-3B-SFT | 33.77 | 22.50 | 29.19 | 29.77 | 48.72 |
| **GeoSense-4B (Ours)** | **37.36** | 36.56 | 35.41 | 31.39 | 44.89 |

feature modulation

### 5.4. Capability on Embodied Task

To further examine how our model performs in more realistic embodied settings, we evaluate it on the recent Quantiphy-validation (Puyin et al., 2025) benchmark showed in Table 5. Compared with both general-purpose and specialised models of a similar scale, our model achieves the best overall performance. Owing to the relatively balanced composition of 2D and 3D tasks in this benchmark, the results suggest that our approach is able to maintain a reasonable trade-off between 2D perception and

3D geometric reasoning, which is consistent with our earlier analysis.

Although our model does not reach the absolute state-of-the-art in all four sub-scenarios, it consistently ranks within the top three for each individual setting. In particular, it attains the second-best results on both 2D and 3D static tasks, which aligns well with our expectations. We also observe that performance in dynamic scenarios is generally lower than in static ones. We conjecture that this gap mainly stems from the distribution bias in our fine-tuning data, which contains fewer samples involving significant motion and viewpoint variation.

Moreover, apart from our method, none of the baseline models can maintain stable rankings across all four categories. This phenomenon suggests that existing training paradigms for multimodal models still struggle to cope with the feature variations introduced by viewpoint changes and object motion in embodied environments. The absence of a clear overall performance breakthrough further highlights the necessity of constructing training data and optimisation strategies that are more specifically designed for embodied tasks.

### 5.5. Inference Efficiency

GeoSense introduces additional computation only when the geometry pathway is triggered. On the geometry-heavy VSI-Bench, about 36% of samples activate 3D extraction. This yields an amortized latency of 895 ms/sample on an A100 GPU, compared with 852 ms/sample for the 2D-only baseline and 950 ms/sample for always-on rigid geometry fusion. Non-triggered samples incur only 0.5% additional FLOPs, while triggered 3D-dependent samples take 1008 ms/sample due to geometry extraction and re-inference. This gap reflects the unavoidable cost of invoking an external 3D encoder and running the second-pass reasoning stage. However, because the trigger is inactive for the majority of samples, this cost is averaged over both 2D-only and 3D-dependent queries rather than being imposed on every input. By selectively paying the cost of 3D reasoning only when geometry is likely to affect the answer, GeoSense improves inference efficiency compared with always-on geometry fusion while retaining access to geometric cues for 3D-dependent queries.

### 5.6. Case Study

We visualize representative model outputs in Figure 5 to analyze the internalized perception of geometric necessity. Beyond the typical samples annotated for 3D reasoning in the training set, such as distance analysis and object dimension estimation, a particularly compelling result emerged in the query "How many cats are in the image?" ( Figure 5 (a)). Here, the model explicitly requested geometric auxiliary fea-

tures to effectively distinguish between the actual kitten and its mirror reflection, indicating that it has learned to leverage 3D geometric features to discern the spatial attributes of objects. Furthermore, the model utilized geometric cues to filter out interference from occlusion, successfully avoiding the misclassification of the object as a chair ( Figure 5 (b)).

Regarding general visual tasks, the model predominantly learned to suppress geometric feature activation. Notably, in specific simple directional queries that ostensibly appear to require geometric insight, the model derived the correct answer via semantic and commonsense reasoning instead. For instance, in Figure 5 (f), the spatial relationship could be inferred solely through color correspondence, rendering explicit geometric input unnecessary. Collectively, these examples corroborate that the model has acquired the capability to adaptively perceive the necessity of spatial information based on specific task demands and available context.

## 6. Conclusion

In this paper, we introduced *GeoSense*, a framework that enables MLLMs to autonomously perceive and act upon geometric necessity. Our core contribution is a model-adaptive training pipeline that internalizes spatial awareness by extracting empirical priors directly from the model's own inference discrepancies. It treats geometric features as an on-demand resource, invoking them only when standard 2D perception is insufficient. Empowered by this data-centric approach, *GeoSense* achieves a superior balance: it establishes new SOTA performance on spatial reasoning benchmarks, while strictly preserving general multimodal reasoning capabilities.

**Limitations and Future Work.** GeoSense is still limited by the quality and granularity of the external geometry encoder. In dense or multi-scene inputs, VGGT may produce excessive geometry tokens, increasing the risk of token overload and noisy spatial cues. In wide-angle or discrete multi-view settings, the extracted geometry can also be less reliable, which may weaken the benefit of adaptive triggering. Future work will explore region-localized geometry extraction, token compression modules, and more diverse 3D representations (*e.g.*, depth maps or point clouds), while further optimizing the trigger mechanism for efficient deployment on resource-constrained devices.

## Acknowledgements

This work was partially supported by New Generation Artificial Intelligence-National Science and Technology Major Projection(2025ZD0123100) and by The National Natural Science Foundation of China (NSFC) under no. 62573399 and U25A20530.

## Impact Statement

This work aims to improve the spatial reasoning ability of multimodal large language models by enabling them to selectively invoke geometric information. Potential positive impacts include more reliable embodied AI, robotics, and assistive perception systems. Potential risks include over-reliance on model-generated spatial judgments in safety-critical settings. We encourage careful validation before deployment in real-world decision-making scenarios.

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
