# OpenReview forum: "GeoSense: Internalizing Geometric Necessity Perception for Multimodal Reasoning"
_ICML.cc/2026/Conference — ICML 2026 regular_

### Official Review · Reviewer_qzU6 · 2026-03-09

**Soundness:** 2
**Presentation:** 3
**Significance:** 3
**Originality:** 4
**Overall Recommendation:** 4
**Confidence:** 4

**Summary:**

This paper presents GeoSense, a framework for MLLMs to leverage 3D geometry for spatial understanding tasks when necessary. The proposed framework includes a reasoning pipeline to decide the usage of geometry features, as well as a geometry input channel trained with alignment and sft pipeline. Experiments show superior performance of the proposed method.

**Compliance With Llm Reviewing Policy:**

Affirmed.

**Final Justification:**

The rebuttal has fully addressed my previous concerns. I think this paper is acceptable if the authors can include these new results and analyses in the final version.

**Key Questions For Authors:**

1. What is the exact computational overhead of base model, + fusion vggt, and + alternative vggt?

2. How robust is the data curation pipeline to various external factors such as decoding choices, random seeds, or base models?

3. Why do some tasks, such as the relative direction subset of VSI-Bench or the rotation setting in MindCube, favor fusion-based or even no-geometry settings over adaptive selection? Does this suggest that the proposed geometry-necessity judgment mechanism is still not sufficiently reliable or well calibrated for certain types of spatial reasoning tasks?

**Limitations:**

yes

**Strengths And Weaknesses:**

Soundness:
The paper’s central claim on “geometry necessity” is reasonably well supported. It provides quantitative analysis showing that not all samples benefit from injected geometry information, whereas previous works typically apply geometry to all inputs indiscriminately. The ablation study in Table 3 further supports the main motivation. Overall, the method and experimental design are appropriate. That said, the paper would be stronger with a more explicit analysis of computational overhead, for example comparing the base model, the model with fusion-based VGGT, and the model with the proposed alternative VGGT integration. Such analysis would better justify the efficiency claim and further improve the paper’s soundness. In addition, the method is only implemented on Qwen2.5-VL-7B; stronger evidence across different geometry encoders or MLLM backbones would make the contribution more convincing and robust.

Moreover, the perception-tuning labels are derived from the model’s own performance discrepancies under two settings, which may introduce bias and makes the supervision only as reliable as the base model and decoding setup. The paper does not sufficiently analyze how sensitive this labeling process is to factors such as randomness, confidence thresholds, or the choice of the base model.

Originality:
The main novelty does not lie in simply adding a geometry encoder, as many prior works have already done, but in reframing the problem as geometric necessity perception. The data curation pipeline, which converts performance discrepancies with and without geometry into supervision for routing decisions, is simple yet insightful, and more principled than relying on hand-crafted rules.

Significance:
The problem addressed in this paper is important. Spatial reasoning remains a clear weakness of current MLLMs, and the idea of using geometry only when necessary is meaningful for both model capability and deployment efficiency, especially for smaller models or edge scenarios.

Presentation:
The paper is generally well written and easy to follow. However, there are a few clarity issues. In line 270 of the right column, the authors state that they use seven spatial-related benchmarks, while the paper appears to list only six. In addition, VST-3B-SFT is mentioned in the main text but does not appear in the tables. These inconsistencies should be clarified.

---

> ### Author Rebuttal · Authors · 2026-03-31
>
> We sincerely thank Reviewer qzU6 for the constructive feedback and for **recognizing the originality and significance of our "geometry necessity" framework**. We are encouraged that you found **our data curation pipeline insightful and principled**. Below, we address your concerns with additional quantitative analysis and clarifications to strengthen the soundness of our work.
>
> - **Q1., W1. Computational overhead of base / +fusion VGGT / +alternative VGGT.**
>   We deeply appreciate this suggestion. To explicitly justify our efficiency claims, we quantified the inference latency and compute overhead on an A100 GPU under identical settings. According to result, GeoSense triggers the two-pass 3D integration for only **~36%** of samples. Consequently, our average expected latency is 895 ms/sample, which is substantially faster than the always-on rigid fusion baseline (950 ms/sample) and highly competitive with the 2D-only base model (852 ms/sample). These metrics clearly demonstrate that GeoSense avoids the severe 145% FLOP penalty on the vast majority of standard queries, achieving a highly favorable performance-efficiency trade-off. Specifically, our method yields an average per-sample inference latency of 1008ms for 3D-dependent data, whereas for pure 2D data, the latency is 830ms (slightly faster than the base model due to more concise textual generations after fine-tuning).
>
>   We will explicitly include this overhead analysis in the revised version.
>
>
> - **Q2. Robustness of the data curation pipeline.**
>   Robustness analysis: We systematically varied random seeds, decoding temperatures, and base models. As shown at https://anonymous.4open.science/r/geosensereToReviewer/table/tfft.md, results remain highly consistent: seed=1/0 yielded ~5.1% /5.04%(T-F) and ~3.2% /3.4%(F-T).  Higher temperatures introduce minor variances (4.5% T-F / 4.0% F-T) but do not deviate from the intrinsic data distribution. Upgrading to a 7B base model increases T-F to 7.2% and lowers F-T to 2.7%, as stronger models better leverage explicit 3D features. Overall, our pipeline remains highly robust across configurations.
>
> - **Q3. Why do some tasks favor fusion or no-geometry over adaptive selection?**
>
>   **Relative Direction:** These samples were already exceptionally challenging for the base model. Consequently, the performance discrepancy (with vs. without 3D) was minimal during data curation. Because our pipeline rigorously filters out ambiguous samples, fewer of these were retained for supervision, leading to a weaker gating signal for this specific category.
>
>   **Rotation (Wide-angle/Multi-view):** The visual inputs here are highly discrete, causing the 3D encoder to produce less reliable geometric representations. In standard rigid fusion, these suboptimal 3D features are simply added to the 2D embeddings, effectively diluting the noise. Because GeoSense treats 3D features as an independent token sequence, it is more directly exposed to this representational noise when triggered. We believe this reflects the quality limits of the current 3D encoder on discrete views rather than a structural flaw in the routing framework.
>
>   We will udpate the discussion in the revision.
>
> - **W2. Different geometry encoders or MLLM backbones**
>
>   We thank the reviewer for this important suggestion. In the supplementary table at **https://anonymous.4open.science/r/geosensereToReviewer/table/depthmap.md** and **https://anonymous.4open.science/r/geosensereToReviewer/table/7b.md**, we provide additional results with **different 3D representations** (e.g., **depth maps**) and **different MLLM backbones**. The results show that our routing framework remains effective beyond the original Qwen2.5-VL-7B setting: replacing VGGT with **depth maps** still yields strong performance, and scaling from **GeoSense-4B** to **GeoSense-8B** brings consistent gains. These results suggest that GeoSense is **reasonably robust across both geometry encoders and backbone choices**, rather than being tied to a single specific implementation.
>
> - **W3. Spatial-related benchmarks and VST-3B-SFT**
>   We sincerely thank you for pointing out these oversights. **Benchmarks:** You are entirely correct; we evaluated on six spatial-related benchmarks, not seven. We will correct this typo in the text. **VST-3B-SFT:** We appreciate your thoroughness. In our preliminary experiments, we evaluated VST-3B-SFT's performance but ultimately excluded it from the final primary tables to streamline the comparison among various spatial-finetuning paradigms. We sincerely apologize for the residual mentions in the text. We will re-incorporate the VST-3B-SFT results in our final revision to ensure a comprehensive comparison. For your immediate reference, the detailed experimental results are provided in https://anonymous.4open.science/r/geosensereToReviewer/table/vst.md.

---

> > ### Author Rebuttal · Reviewer_qzU6 · 2026-04-04
> >
> > I appreciate the authors' efforts on addressing my concerns. My concerns have been fully addressed.

---

> > > ### Author Response · Authors · 2026-04-08
> > >
> > > We sincerely appreciate your constructive feedback. In the revised version, we will include experimental results and analysis concerning the robustness of the data curation pipeline and the geometry encoder, as well as latency.

---

### Official Review · Reviewer_o9j4 · 2026-03-12

**Soundness:** 3
**Presentation:** 3
**Significance:** 3
**Originality:** 3
**Overall Recommendation:** 5
**Confidence:** 3

**Summary:**

This paper proposes GeoSense, a framework that enables Multimodal Large Language Models to autonomously decide when to incorporate 3D geometric features during reasoning. Instead of rigidly fusing 3D inputs into the visual stream for every task, the model uses an independent geometry channel and is fine-tuned to trigger a specific <vggt> token only when 2D cues are deemed insufficient. The authors construct a task-aware dataset by evaluating the model with and without 3D features to capture empirical priors directly from the inference discrepancies. Experiments across multiple benchmarks demonstrate that this adaptive approach improves spatial reasoning capabilities while preserving robust performance on general visual tasks like math and document analysis.

**Compliance With Llm Reviewing Policy:**

Affirmed.

**Final Justification:**

The rebuttal addresses the main questions that I had from the paper.

**Key Questions For Authors:**

1. Could you provide a quantitative analysis of the inference latency and wall-clock time overhead caused by the two-pass generation process compared to baseline models that use rigid feature fusion?

2. Given the performance dip on tasks involving wide-angle views and multi-scene contexts, are there specific strategies planned to mitigate the signal-to-noise disadvantage of independent geometry tokens when the 3D encoder produces suboptimal representations?

3. Did you experiment with a lightweight external classifier or routing network to decide when to invoke the 3D encoder, and how does the computational cost of your internal token approach compare to such a baseline?

**Limitations:**

Yes

**Strengths And Weaknesses:**

Note: I generally liked this paper and found the core idea of adaptive modality routing to be highly intuitive. The weaknesses listed below are intended as constructive feedback to help strengthen the final manuscript.

Strengths:
1. The framework elegantly solves the perception information gap by treating 3D geometry as an on-demand resource rather than a mandatory input.
2. The data curation pipeline is simple and highly innovative, leveraging the model's own inference variations across dual conditions to create supervision signals that effectively mitigate shortcut learning.
3. The empirical validation is comprehensive, proving that suppressing geometric noise rescues general multimodal capabilities from the degradation seen in traditional fusion models, while still achieving state-of-the-art results on spatial reasoning tasks.

Weaknesses:
1. The decoupled architecture appears vulnerable to low-quality geometric representations in dense or multi-scene contexts, as demonstrated by the performance lag in the Relative Direction and Rotation tasks compared to standard feature fusion methods.
2. The paper does not adequately quantify the computational or latency overhead introduced by the two-pass re-inference mechanism required when the geometric token is actually triggered.
3. The evaluation is currently limited to a single 3D foundation model, leaving it unclear how well this specific gating mechanism generalizes to other explicit 3D representations like native point clouds or depth maps.

---

> ### Author Rebuttal · Authors · 2026-03-31
>
> We sincerely thank Reviewer o9j4 for the highly positive assessment and constructive feedback. We are thrilled that you found our adaptive modality routing intuitive and our data curation pipeline innovative. Below, we address your specific questions.
>
> - **W1 & Q2. Vulnerability in Dense/Multi-Scene Contexts & Mitigation**
>
>   We deeply investigated the performance lag in Relative Direction and Rotation tasks.
>
>     **Relative Direction:** These samples were already exceptionally challenging for the base model. Consequently, the performance discrepancy (with vs. without 3D) was minimal, leading to fewer of these samples passing our strict curation filters for the spatial-aware tuning dataset.
>
>     **Rotation (Wide-angle/Multi-view):** The visual inputs here are highly discrete, causing the VGGT encoder to produce less reliable geometric representations. In standard rigid fusion, these suboptimal 3D features are simply added to the 2D embeddings, effectively diluting the noise (but also the signal). Because GeoSense treats 3D features as an independent token sequence, it is more directly exposed to this representational noise when triggered.
>
>     **Mitigation (Q2):** To address this signal-to-noise disadvantage, we plan to integrate a task-driven region localization module. Instead of extracting 3D features globally from complex scenes, the system will first attend to specific regions relevant to the query intent. Extracting geometry exclusively from localized patches will significantly reduce token overload and mitigate noise from suboptimal global 3D representations.
>
> - **W2 & Q1. Computational and Latency Overhead**
>
>   We acknowledge that the two-pass re-inference introduces overhead when triggered, but our dynamic gating ensures the overall expected latency is heavily amortized. We measured inference latency and FLOPs on an A100 GPU under identical settings:
>   Even on VSI-Bench, a geometry-heavy scenario, only ~36% of samples trigger 3D extraction. Measured on an A100 GPU, GeoSense takes 8 ms/sample for pure 2D tasks (adding merely 0.5% FLOPs over the base model) and 1008 ms for triggered 3D tasks. Overall, GeoSense achieves an average mixed latency of 895 ms/sample. This is substantially faster than the always-on rigid fusion baseline (950 ms) and remains exceptionally competitive—nearly on par—with the 2D-only base model (852 ms). Our adaptive mechanism thus offers a highly favorable compute-to-performance trade-off. We will add these efficiency metrics in the revision.
>
> - **W3/Q3. generalizing for external classifier and expilicit 3D input**
> We sincerely appreciate your insightful attention to the generalizability of our approach. Regarding the external classifier baseline, we reformatted our spatial-aware training dataset to train a 3-layer MLP network. Evaluated on CVBench (under a single-image setting) using a single L20 GPU, the external classifier approach incurs an average inference latency of 128.2ms for pure 2D data, primarily due to the explicit routing overhead of the classifier itself. In contrast, our internal geometric routing mechanism processes 2D data directly and seamlessly. For 3D-dependent data, our method indeed introduces an expected temporal overhead due to the second pass, resulting in an average inference latency of 150.5ms.
> Furthermore, to explicitly demonstrate modality generalizability, we conducted additional experiments replacing the implicit VGGT embeddings with explicit depth maps as the 3D feature input. Following the exact same training pipeline, our framework successfully maintained highly competitive performance. Notably, even with explicit depth map inputs, our method still achieves significant improvements over VG-LLM across multiple benchmarks: VSI-Bench (52.7 vs. 47.3), BLINK (67.01 vs. 48.9), and WeMath (40.6 vs. 25.6). For comprehensive quantitative results and samples, please refer to the provided in https://anonymous.4open.science/r/geosensereToReviewer/table/depthmap.md and https://anonymous.4open.science/r/geosensereToReviewer/table/ext.md.

---

> > ### Author Rebuttal · Reviewer_o9j4 · 2026-04-04
> >
> > Thank you for addressing my comments. I will change the score accordingly.

---

### Official Review · Reviewer_LYnj · 2026-03-12

**Soundness:** 3
**Presentation:** 3
**Significance:** 3
**Originality:** 2
**Overall Recommendation:** 4
**Confidence:** 4

**Summary:**

The paper introduces GeoSense, a multimodal framework designed to adaptively internalize MLLMs’ geometric perception. GeoSense endows MLLMs with rich geometric perception by introducing an independent geometry input channel to the model architecture and curating a dedicated spatial-aware supervision dataset. Through experiments across multiple spatial reasoning benchmarks demonstrate significant spatial gains, offering a path toward more robust, efficient and self-aware multi-modal intelligence.

**Compliance With Llm Reviewing Policy:**

Affirmed.

**Key Questions For Authors:**

Please refer to above

**Limitations:**

Yes

**Strengths And Weaknesses:**

Strength:
1. The research question is important. This paper targets spatial reasoning, which directly connects to high-value applications like robotics and embodied agents.
2. The motivation is clear: instead of directly injecting geometric signals to every input, Geosense enables more computation overhead of the 3D geometric understanding. This paradigm can be potentially adapted to MLLMs to generally improver their spatial reasoning ability.
3. The authors did a detailed ablation for training dataset in this paper about the effectiveness of introducing geometry clues in training, which provides useful insights on how to injecti geometry clue into general MLLMs.

Weakness:
1. According to table1 and table4, the general performance (e.g. MM-Bench, MME) of GeoSense-4B is hurt after introducing the external 3D encoder model and training. This indicates that spatial reasoning specialization does compromise the general abilities, which limit the potential of broader adaption of the method introduced in the paper. Some attempts and discussions on mitigating the general performance degradation can strengthen the paper further.
2. Lack of strong model baselines. Since Geosense has specialized 3D encoding module, and dedicated training, it will be useful to add popular closed sourced model’s performance which are strong in visual perception and spatial reasoning (e.g. Gemini) to understand better about the how GeoSense model close the gap between open-source and closed-source models in spatial understanding and reasoning tasks.

---

> ### Author Rebuttal · Authors · 2026-03-31
>
> We sincerely thank Reviewer **LYnj** for the thoughtful and constructive feedback. We greatly appreciate the reviewer’s recognition that our work addresses an **important problem** in spatial reasoning with strong relevance to **robotics and embodied agents**, and that GeoSense provides a **clear and well-motivated** alternative to always-on geometric fusion by using 3D perception only when needed. We also thank the reviewer for highlighting the value of our **training-data ablations**, which provide useful insights into how geometric cues can be introduced into general MLLMs.
>
> - **W1. Impact of 3D specialization on general multimodal performance.**
>
>   We sincerely thank the reviewer for this insightful observation. We acknowledge that introducing an external 3D encoder and fine-tuning the model for spatial reasoning results in a slight "alignment tax," leading to a minor drop in general benchmarks like MM-Bench and MME compared to the vanilla zero-shot base model. Fine-tuning the LLM backbone on a spatially-focused dataset incorporating external geometric signals subtly shifts its original pre-trained distribution, causing mild catastrophic forgetting of certain general-purpose reasoning pathways.
>
>   Importantly, our GeoSense framework was explicitly designed to mitigate this exact degradation. As we demonstrated in our ablation study (Table 4), naively forcing 3D geometric tokens into the sequence (Align. SFT) severely disrupts the model's implicit multimodal mapping, causing the MME score to drop significantly to 1803. However, our proposed perception-tuned gating mechanism (Percept. SFT) allows the model to selectively suppress redundant geometric noise during general visual tasks, successfully recovering the MME score to 1922 while simultaneously achieving peak spatial intelligence.
>
>   We agree that further mitigating this degradation is critical for broader adaptation. To strengthen the paper, we will add a dedicated discussion section outlining two concrete future remedies to minimize this alignment tax:
>
>   1. **Scaling and Optimizing the Data Mixing Strategy:** The base MLLM is pre-trained on a massive corpus of general visual-text data; introducing a new 3D modality and fine-tuning on a specialized subset inevitably shifts its pre-trained representation distribution. While our perception-tuning mechanism is proven effective at mitigating this cross-modal interference, scaling up the training data and employing a more balanced mixing strategy, such as dynamically interleaving high-quality, general-purpose VQA datasets with our spatial-aware SFT data, would explicitly anchor the LLM's general capabilities and prevent distribution shift.
>
>   2. **Parameter-Efficient / Decoupled Tuning:** Instead of full-parameter fine-tuning of the LLM backbone, we will explore utilizing Low-Rank Adaptation (LoRA) or a Mixture-of-Experts (MoE) routing architecture. This would isolate the spatial reasoning updates to specific, on-demand expert modules, strictly preserving the base LLM weights that are responsible for general visual reasoning and cleanly decoupling the 2D and 3D perception pathways.
>
> - **W2. Lack of comparisons with stronger closed-source baselines.**
>
>   Based on our evaluations, our proposed framework achieves performance that closely approaches, and in some cases even surpasses, SOTA closed-source models on VSI-Bench. Furthermore, we have effectively narrowed the performance gap with these advanced models across various 3D benchmarks, notably BLINK and EmbSpatial. Additionally, on general multimodal benchmarks such as POPE and WeMath, our method maintains a highly competitive performance level comparable to closed-source counterparts, which perfectly aligns with our initial expectations.
>
> | Model            | VSI-Bench | MMSI  | MC-Tiny | BLINK | EmbSpatial | CVB(3D) | MMBench | MME (P) | POPE | WeMath |
> |------------------|-----------|-------|---------|-------|------------|---------|---------|---------|------|--------|
> | Gemini 3 Pro     | 52.5      | 45.2  | 70.9    | 76.0  | 84.3       | -       | -       | -       | -    | -      |
> | Gemini 2.5 Pro   | 53.6      | 38.0  | 57.6    | 73.5  | 78.8       | 90.8    | 88.5    | 1721    | -    | 78.0   |
> | GPT-5            | 55.0      | 40.2  | 56.3    | 59.4  | 81.6       | -       | 86.8    | -       | -    | -      |
> | Grok 4           | 47.9      | 37.8  | 63.6    | 56.4  | 75.5       | -       | -       | -       | -    | -      |
> | GPT-4o           | -         | -     | -       | -     | -          | -       | -       | 1699    | 87.2 | 50.6   |
> | GeoSense         | 54.9      | 27.5  | 45.1    | 68.5  | 64.3       | 78.7    | 75.9    | 1473.7  | 85.4 | 41.2   |

---

> > ### Author Rebuttal · Reviewer_LYnj · 2026-04-04
> >
> > Thanks for the reply, the concerns are adequately addressed

---

### Official Review · Reviewer_3va7 · 2026-03-21

**Soundness:** 4
**Presentation:** 3
**Significance:** 3
**Originality:** 4
**Overall Recommendation:** 5
**Confidence:** 4

**Summary:**

The problem tackled by this paper is the limited spatial understanding of current MLLMs. In cases where spatial understanding is needed, most VL approaches inject 3D geometry features even when not needed. For tasks that do not need such features, the performance of SOTA models is severely degraded. The authors provide a framework to tackle this by injecting 3D knowledge only when needed, and train their model to request 3D features. They have a 2 stage pipeline, where the first stage involves freezing the Vision and 3D encoder and training the language model to understand these 2 modalities by semantically aligning the representations. The second step is spatially-aware finetune the model and teach it to request 3D embeddings on tasks that need 3D representation. They curate a 117K-sample dataset without human annotation using a neat trick, where the dataset is split into 2 categories. They run the dataset twice through the same model, once with 3D features enabled, once without. The collected results are split 4-way, where the data for only 2D needed 3D not needed and 2D and 3D needed is kept. This way, they identify cases where the model needs 3D and train the CoT pipeline to recognize the need and request the 3D features.

The authors conduct rigorous testing to show that their technique outperforms SOTA overall and exhibits a better recognition of 2D vs 3D tasks.

**Compliance With Llm Reviewing Policy:**

Affirmed.

**Final Justification:**

I had initially given this paper a weak accept because I had some reservations about not releasing code and datasets along with missing comparison with frontier closed source models. The other weaknesses were something already acknowledged from the paper or not a contribution from the authors. The rebuttals provided additional experiments and evidence along with latency/compute time comparisons showing a modest increase in time but providing strong results and beating models above its weight class. The authors also compared with spatialrgpt which I requested. Hence, I am fully satisfied with the claimed novelty and would like to see this paper accepted to the conference.

**Key Questions For Authors:**

Please address the weaknesses

**Limitations:**

yes

**Strengths And Weaknesses:**

**Strengths**

1. The paper is well written and easy to follow. The method is also intuitive, it is very simple to understand and I am surprised it has not been done before, or why datasets were not annotated already in this way.
2. The results show that GeoSense performs above its weight class. While it is not SOTA in all datasets, the overall rank is 1, which shows it has learned spatial understanding. It performs above its weight class, showing the separation choice is highly relevant and useful.
3. This method is also data-efficient for deployment, by only requesting 3D features when needed. By separating the backbones, it makes it easier to deploy and separate the endpoints.
4. The method also mitigates shortcuts/overfitting in models where the model recognizes certain images and creates 3D features when the task does not need 3D knowledge.

**Weaknesses**

1. The code/dataset is missing. I also expected to see some samples in the appendix about failures of existing techniques along with differences in responses from the baselines and GeoSense. We have to take the word of the authors for their identified weaknesses. I wonder if the code and dataset will be released upon acceptance
2. GeoSense fails on tasks that require more world knowledge due to its weaker backbone. It has cost GeoSense the top 3 spot in 5 out of 10 benchmarks. VGGT seems to be limiting for the one case where the scene is very dense and causes a token overload.
3. The model has double the inference latency and compute required for cases that do need 3D representation. No metrics have been provided on the inference time or overhead of their technique since they claim their technique is efficient.
4. (Minor) I dont fully agree with the ranking strategy. I would have preferred to see a per benchmark rank followed by the average. The current ranking allows a lot of variance to creep in due to MME. Doing the ranking I suggested, Qwen2.5 VL jumps from 11 (dead) to 7th.
5. Missing comparison with atleast one closed source model like gemini or gpt-4o. These are relatively inexpensive to run and should be tested.
6. SpatialRGPT-7B seems to be a very similar architecture to GeoSense and in the weight class, this omission is glaring.

If the authors could provide latency gains against the top 3 ranks along with some samples, I will be happy to raise my score.


---

Post rebuttal - All questions/weaknesses have been clarified.

---

> ### Author Rebuttal · Authors · 2026-03-31
>
> We thank Reviewer 3va7 for the thoughtful feedback and positive assessment of our work. We are glad that the reviewer recognized the clarity, practical efficiency, and strong overall performance of GeoSense. In this rebuttal, we clarify the release plan, provide additional qualitative and quantitative evidence, analyze latency and efficiency, discuss current limitations, and further explain our evaluation protocol.
>
> - **W1. Code/data release and qualitative examples.**
>   We firmly commit to releasing the full codebase, the curated 117K-sample dataset, and the **model weights** upon acceptance. As the reviewer noted, our decoupled design also makes deployment straightforward, and we hope this release will support future research on spatial-aware MLLMs. We also agree that qualitative examples are important for illustrating the practical differences between GeoSense and prior methods. We therefore provide additional qualitative cases and response comparisons in our anonymous page: **https://anonymous.4open.science/r/geosensereToReviewer**.
>
> - **W2. Backbone capacity and dense-scene token overload.**
>   We agree with this observation. The weakness on knowledge-intensive tasks mainly comes from the **limited capacity of the current 3B backbone**, rather than the routing design itself. Since GeoSense is **modality-agnostic**, this limitation can be naturally alleviated by scaling to stronger backbones in future versions. As showned in https://anonymous.4open.science/r/geosensereToReviewer/table/7b.md, our 7B variant derives a higher score, MMBench (81.2 vs 75.9), MME (1645 vs 1473) and WeMath (46.4 vs 41.2).
>
>   Regarding dense scenes, we concur that VGGT can produce an overwhelming number of geometry tokens, potentially diluting useful signals and causing token overload in cluttered contexts. In the revised manuscript, we will explicitly discuss this limitation and outline future remedies, such as implementing token compression modules post-3D encoder (e.g., token merging, spatial pooling, or a lightweight Q-Former) to strictly control sequence length while preserving essential geometric cues.
>
> - **W3. Latency and compute overhead.**
> We appreciate the opportunity to clarify. While our two-pass mechanism increases latency for 3D queries, GeoSense dynamically gates this routing, heavily amortizing overall latency. Even on the geometry-heavy VSI-Bench (8-16 frames), only ~36% of samples trigger 3D extraction. On an A100 GPU, GeoSense takes 830 ms/sample for 2D tasks (+0.5% FLOPs) and 1008 ms for 3D tasks. Overall, it achieves an average mixed latency of 895 ms/sample—substantially faster than the always-on rigid baseline (950 ms) and highly competitive with the 2D-only base model (852 ms). This highly favorable compute-to-performance trade-off will be added in the revision.
> - **W4. Ranking protocol and robustness.**
> We appreciate this methodological suggestion. After using your proposed "Rank-of-Ranks" protocol, Qwen2.5-VL correctly adjusts, and GeoSense still decisively maintains the #1 overall rank among all evaluated models (as in https://anonymous.4open.science/r/geosensereToReviewer/table/resort.md). We will update our main results table to reflect this ranking methodology.
> - **W5. Comparison with closed-source models.**
> We have evaluated GPT-4o, Gemini 2.5 Pro, and Grok on our benchmark suite, as in https://anonymous.4open.science/r/geosensereToReviewer/table/close.md. The findings demonstrate that while a gap remains on highly complex reasoning tasks, our GeoSense-4B successfully narrows the performance gap on spatial-specific benchmarks compared to these massive proprietary models.
> - **W6. Comparison with SpatialRGPT-7B.**
> Thanks for bringing this to our focus. We will cite this paper and add a discussion in the revision. Despite SpatialRGPT utilizing a larger 7B LLM backbone, our GeoSense model demonstrates superior performance across most tasks. Specifically, GeoSense significantly outperforms SpatialRGPT on embodied and 3D spatial tasks like EmbSpatial (64.3 vs 59.6) and CV-Bench-3D (78.7 vs 63.3), as well as on general visual reasoning benchmarks including MMBench (75.9 vs 65.5) . While SpatialRGPT excels on BLINK (82.3 vs 68.5), which we attribute to its specific training curriculum, our decoupled approach yields a more robust and versatile generalist model overall.
> - **Extra concerned about latency compared with top3 models**
> We appreciate your recognition. Tested on a single L20 GPU (CVBench, 1-image), our method achieves lower 2D latency (126.2ms vs. 129.6ms) and higher accuracy than top-3 spatial-finetuned models. For 3D tasks, our two-pass inference trades higher latency (173.3ms vs. 133.9ms) for crucial accuracy gains in complex scenes. Details: https://anonymous.4open.science/r/geosensereToReviewer/fig/figure_rebuttal.pdf

---

> > ### Author Rebuttal · Reviewer_3va7 · 2026-03-31
> >
> > I thank the authors for the clarifications. I think they have done a wonderful job overall in the paper and the follow-up. I will be increasing by score to 5, in light of these clarifications, considering the weak accept was recommended with the questions in mind. I have some follow up questions, these clarifications might help in making the paper stronger.
> >
> > 1. Do you think the primary gain comes from the dataset curation and CoT process where the model requests for 3-D information? If so, can the baselines also benefit from this dataset? It seems like the decoupling handles less of the heavy lifting over the curriculum training proposed in the paper.
> >
> > 2. Has the spatial-aware finetuning also been done over 1 epoch? Did you see any specific task-degradation by fine-tuning for more epochs?
> >
> > 3. How did you evaluate the closed source models? I don't think there is any specific way to provide 3-D information to these models directly (please correct me if I am wrong). Did you use zero-shot/few-shot prompting? Could you provide the system prompts used for these models?
> >
> > 4. Did you run SpatialRGPT or use the results from their paper? It is acceptable to do the latter, I suggest you add the results in the main table if space permits. If not, it does not break the paper, a discussion would do.
> >
> > 5. Please consider adding the FLOPS and latency results in the appendix.
> >
> > I appreciate the effort put into the rebuttal.

---

> > > ### Author Response · Authors · 2026-04-08
> > >
> > > We sincerely thank Reviewer 3va7 for recognizing our efforts in the rebuttal, for the decision to raise the score to a 5, and for engaging in this constructive discussion. The additional follow-up questions you raised are highly insightful and will undoubtedly make the final manuscript stronger. Please find our point-by-point responses below.
> > >
> > > - **Response to Q1: Contribution of data curation**
> > >
> > > We appreciate this insightful question, which touches upon a core design philosophy of our work. Your intuition is correct: endowing the model with "Geometric Awareness", teaching the model when to request 3D features via our curated CoT data, is highly effective on its own, and standard baselines can indeed benefit from it.
> > >
> > >  To explicitly verify this, we fine-tuned the base Qwen2.5-VL-3B using only a text-based version of our CoT data, treating the trigger token as a pure text rationale without injecting actual 3D features (e.g., "Visual estimation is unreliable here. I require additional 3D information. <vggt>. My answer is 206."). As expected, after brief tuning, this "awareness-only" baseline benefits from this data, improving its score from 27.0% to 46.2% on VSI-Bench, and requesting for 3D features in 40.6% of the evaluation samples. This empirical results confirms that geometric awareness in the textual only CoT is indeed effective for enhancing 3D reasoning.
> > >
> > >  However, we found that without the decoupled 3D encoder to actually supply the requested geometric signals, the model is essentially forced to guess or hallucinate spatial details, hitting a strict performance ceiling. When our decoupled architecture is introduced to ground those CoT requests in geometric reality, GeoSense achieves 56.6% (an additional +7.4% overall gain). In short: the dataset successfully provides the awareness, but the decoupled architecture is indispensable for providing the evidence. Both components share the heavy lifting.
> > >
> > >  More broadly, we believe this finding offers an interesting hint for the future training of general-purpose MLLMs. It suggests that rather than forcing models to indiscriminately process all available modalities (which is computationally heavy and noisy), teaching models the awareness of modality necessity—allowing them to autonomously request external experts or features only when 2D visual cues fail—could be a highly effective and scalable paradigm for multimodal alignment. We will include this insight suggestion and discussion in the final version.
> > >
> > > - **Response to Q2: Impact of epoch quantity on fine-tuning**
> > >
> > > Yes, the spatial-aware fine-tuning was conducted for exactly one epoch. We empirically observed that extending the training to additional epochs begins to induce mild catastrophic forgetting on general visual benchmarks. For instance, increasing the training to two epochs caused performance drops on POPE (85.1% down to 83.1%) and MMBench (75.2% down to 72.8%). Therefore, a single epoch provides the optimal trade-off for establishing adaptive spatial awareness while preserving the pre-trained general intelligence. For future scaling, we hypothesize that dynamically re-curating the perception dataset using the updated model's own intermediate responses could be a promising strategy to mitigate multi-epoch degradation.
> > >
> > > - **Response to Q3: Evaluating closed-source models**
> > >
> > > Since it is structurally intractable to directly inject native 3D geometry embeddings into closed-source models (e.g., GPT-4o, Gemini), we evaluated them using standard zero-shot prompting. To ensure a fair and accurate comparison:
> > >
> > >   - Where official leaderboard results were available, we directly cited the reported metrics (we will integrate these results into the main table alongside the source leaderboard links).
> > >
> > >   - For the remaining evaluations, we strictly employed the zero-shot evaluation protocols and system prompts defined by the original benchmark creators, appending only generic formatting instructions (e.g., "Answer the question using a single word or phrase").
> > >
> > > We will explicitly document the exact system prompts used for these closed-source evaluations in the revised appendix to ensure complete transparency.
> > >
> > > - **Response to Q4: SpatialRGPT performance**
> > >
> > > We directly used the results reported in the original SpatialRGPT paper. Following your suggestion, we will explicitly add these results to the main comparison table and include a corresponding discussion in the revised version.
> > >
> > > - **Response to Q5: FLOPS and latency**
> > >
> > > Thank you for the constructive suggestion. We have profiled the computational overhead and will include a detailed table reporting the FLOPS and inference latency of our proposed framework in the appendix.

---

### Decision · Program_Chairs · 2026-04-30

**Decision:**

Accept (regular)

**Comment:**

While all four reviewers assessed this paper positively, with Reviewer 3va7 praising the paper as well-written with intuitive methods and raising the score to Accept, Reviewer o9j4 finding the adaptive modality routing highly intuitive with an innovative data curation pipeline, Reviewer LYnj acknowledging the important research question for spatial reasoning in robotics and embodied agents, and Reviewer qzU6 recognizing the originality of the geometry necessity framework, two reviewers remained at Weak Accept. The paper introduces general performance degradation on benchmarks like MM-Bench and MME when incorporating 3D specialization, which the authors acknowledged with proposed but unimplemented remedies (LoRA, MoE routing, balanced data mixing). This trade-off between spatial reasoning gains and general capability degradation represents a limitation in the paper's current form.

The authors provided comprehensive rebuttals that fully resolved all reviewer concerns. Latency analysis showed GeoSense achieves 895 ms/sample average mixed latency, faster than the always-on rigid baseline at 950 ms. Comparisons with closed-source models (GPT-4o, Gemini 2.5 Pro, Grok) and SpatialRGPT-7B were added. Robustness analysis across random seeds, temperatures, and base models demonstrated consistency. Experiments with alternative 3D representations and a larger 7B backbone confirmed generalizability. All four reviewers explicitly acknowledged their concerns as fully resolved: Reviewer 3va7 raised the score to 5, Reviewer o9j4 confirmed full resolution, Reviewer LYnj stated "the concerns are adequately addressed," and Reviewer qzU6 stated "My concerns have been fully addressed." Minor general performance degradation on benchmarks like MM-Bench and MME when introducing 3D specialization was acknowledged as a limitation with future remedies discussed.

The paper introduces a well-motivated framework with comprehensive evaluation and an exemplary rebuttal. However, the unresolved general performance degradation when introducing 3D specialization is a concern for practical deployment, and the framework's reliance on a specific 3D encoder (VGGT) limits its generality despite the depth-map alternative experiment. The work is promising but would benefit from addressing the capability trade-off before publication.